# New Insight into the Function of Dopamine (DA) during Cd Stress in Duckweed (*Lemna turionifera* 5511)

**DOI:** 10.3390/plants12101996

**Published:** 2023-05-16

**Authors:** Wenqiao Wang, Yunwen Yang, Xu Ma, Yuman He, Qiuting Ren, Yandi Huang, Jing Wang, Ying Xue, Rui Yang, Yuhan Guo, Jinge Sun, Lin Yang, Zhanpeng Sun

**Affiliations:** 1Tianjin Key Laboratory of Animal and Plant Resistance, College of Life Sciences, Tianjin Normal University, Tianjin 300387, China; 1930170110@stu.tjnu.edu.cn (W.W.); 1930170092@stu.tjnu.edu.cn (Y.Y.); 2110170006@stu.tjnu.edu.cn (X.M.); 2030170114@stu.tjnu.edu.cn (Y.H.); 2010170012@stu.tjnu.edu.cn (Q.R.); 2030170118@stu.tjnu.edu.cn (Y.H.); 2030170137@stu.tjnu.edu.cn (J.W.); 1930170202@stu.tjnu.edu.cn (Y.X.); 1930170204@stu.tjnu.edu.cn (R.Y.); 1910170008@stu.tjnu.edu.cn (J.S.); 2Shanghai Key Laboratory of Regulatory Biology, Institute of Biomedical Sciences, School of Life Sciences, East China Normal University, Shanghai 2002141, China; 51251300092@stu.ecnu.edu.cn; 3Faculty of Education, Tianjin Normal University, Tianjin 300387, China

**Keywords:** Dopamine (DA), Cadmium (Cd), duckweed, Ca

## Abstract

Dopamine (DA), a kind of neurotransmitter in animals, has been proven to cause a positive influence on plants during abiotic stress. In the present study, the function of DA on plants under cadmium (Cd) stress was revealed. The yellowing of duckweed leaves under Cd stress could be alleviated by an exogenous DA (10/20/50/100/200 μM) supplement, and 50 μM was the optimal concentration to resist Cd stress by reducing root breakage, restoring photosynthesis and chlorophyll content. In addition, 24 h DA treatment increased Cd content by 1.3 times in duckweed under Cd stress through promoting the influx of Cd^2+^. Furthermore, the gene expression changes study showed that photosynthesis-related genes were up-regulated by DA addition under Cd stress. Additionally, the mechanisms of DA-induced Cd detoxification and accumulation were also investigated; some critical genes, such as vacuolar iron transporter 1 (VIT1), multidrug resistance-associated protein (MRP) and Rubisco, were significantly up-regulated with DA addition under Cd stress. An increase in intracellular Ca^2+^ content and a decrease in Ca^2+^ efflux induced by DA under Cd stress were observed, as well as synchrony with changes in the expression of cyclic nucleotide-gated ion channel 2 (CNGC2), predicting that, in plants, CNGC2 may be an upstream target for DA action and trigger the change of intracellular Ca^2+^ signal. Our results demonstrate that DA supplementation can improve Cd resistance by enhancing duckweed photosynthesis, changing intracellular Ca^2+^ signaling, and enhancing Cd detoxification and accumulation. Interestingly, we found that exposure to Cd reduced endogenous DA content, which is the result of a blocked shikimate acid pathway and decreased expression of the tyrosine aminotransferase (TAT) gene. The function of DA in Cd stress offers a new insight into the application and study of DA to Cd phytoremediation in aquatic systems.

## 1. Introduction

Cadmium (Cd) causes environmental pollution mainly through a series of human activities, including fertilization, industrial activities, and the discharge of sewage sludge. Cd has caused serious problems in both ecosystems and human health [1,2]. After the aquatic system is polluted by Cd, this harmful element will enter the human body through food chains by the drinking of Cd-containing water or consumption of plants irrigated with contaminated water. Cd causes diseases such as Cd nephropathy or itai-itai disease, and has also been proven as a human carcinogen. Furthermore, it could be retained in the human body for a lifetime [1,3]. Therefore, it is of great significance to remediate Cd contamination in aquatic system. Phytoremediation through hyperaccumulator plants is proving to be a profitable option [4]. However, few aquatic plants have strong Cd adsorption and resistance. It is essential to enhance the Cd accumulation and Cd resistance in aquatic plants.

Plants do not have active mobility as animals do, thus their strategies for adjusting and resisting external challenges are more embedded in particular life processes. Studies in recent decades have shown that neurotransmitters are not only essential signaling molecules in animals, but also play an indispensable role in plants during stress. Firstly, glutamate (Glu) plays a role during wound and Cd stress. In 2013, Mousavi et al. found that in *Arabidopsis*, if Glu receptors were mutated, the amount of jasmonate in leaves distal to wounds would decrease. At first, they predicted that glutamate (Glu) would play a role in long-distance transmission [5]. Subsequently, in 2018, Toyota found the long-distance Glu signal response during wound stress, which was accompanied by a calcium (Ca) signal at the same time [6]. Furthermore, in our previous studies, we found that the Glu signal responds during Cd stress in duckweed [7]. In addition, it has been found that inhibitory neurotransmitter GABA plays a part in Cd response [8]. Moreover, acetylcholine reduced the adverse effects of Cd stress in tobacco. However, few studies related to neurotransmitters such as Dopamine (DA) have been studied during Cd stress [9]. DA plays a role as critical excitatory neurotransmitter in the human central nervous system. Its involvement in long-distance signal response to Cd stress in plants needs to be explored.

It has been demonstrated that DA plays an important role in the improvement of the ecological environment, especially in the field of stress resistance research [10]. (i) Both endogenously or exogenously, DA can improve plant tolerance against abiotic stresses to varying degrees, for instance, drought, alkali, and nutrient stress [11,12,13]. (ii) The addition of DA enables plants to resist pests and reduces the threat of pathogens [14]. (iii) DA can alleviate the accumulation of phytotoxicity induced by organic pollutants in water, like Bisphenol A [15]. (iv) DA impacts the expression of many stress-related genes, which emphasizes its role as a multi-regulatory molecule. These facts demonstrate the strong application potential of DA in phytoremediation, as well as in improving resistance in plants. However, studies on the effects of DA addition on Cd tolerance or accumulation in plants under Cd stress have not been reported and, thus, will be addressed in our paper. 

In animals, DA is synthesized in the cytoplasm from L-tyrosine. Once tyrosine enters the neuron, catalyzed by tyrosine hydroxylase in the cytoplasm, it is converted to levodopa. Then, mediated by aromatic L-amino acid decarboxylase, levodopa is decarboxylated to DA rapidly [16]. DA biosynthetic pathways in plants are similar to those in animals. In summary, there are two pathways, both of which require tyrosine as the precursor substance, which is produced by the shikimate acid pathway. The first pathway begins with the tyrosine hydroxylase mediated conversion of tyrosine to L-dopa, followed by dopa decarboxylase to decarboxylate it to DA. The second pathway is in the reverse order of the first. Tyrosine is primarily decarboxylated by tyrosine decarboxylase to produce tyramine, which is then catalyzed by monophenol hydroxylase to produce DA [10,17,18]. The regulation of DA synthesis and metabolism pathways in plants during Cd stress has not yet been investigated. In animals, the DA signal has been fully studied. It mainly consists of two signal pathways: one is the activation of adenylate cyclase (ADCY) and, thus, the production of a cAMP-PKA mediated signal; the other is the activation or inhibition of Phospholipase C (PLC), so that the phosphatidylinositol signal pathway can be regulated. It should be emphasized that the final way of regulation is determined by different stimulation and different type of DA receptor [19,20]. However, in plants, only a few studies about DA signal have been reported. It was predicted that the DA target in plants may be related to cAMP and the cyclic nucleotide-gated ion channel (CNGC), which seems to have some similarities with the DA target in animals [21,22]. Thus, more details of the expression of the DA signal pathway related genes in plants need to be explored.

Duckweed is one of the smallest but fastest growing aquatic plants, and it has been used for the phytoremediation of wastewater, radioactive waste, toxins, and agrochemicals [23,24]. Duckweed is regarded as a safe, environmentally friendly, and economical plant for the treatment of heavy metals. It is capable of adsorbing Cd in polluted water, and yet it is sensitive to the water environment [25]. The growth of duckweed was affected when it was exposed to over 0.5 mg·L^−1^ of Cd [26]. Therefore, the improvement of Cd tolerance and resistance in duckweed is crucial. In this study, *Lemna turionifera* 5511 was used as the material to study the effect of DA treatment on the tolerance and accumulation of Cd in duckweed.

Our study focuses on the responses of duckweed treated with DA to Cd stress. The main objectives are as follows: (i) to investigate the effect of exogenous application of DA on the Cd resistance and enrichment capacity of duckweed under Cd stress; (ii) to describe the molecular mechanisms of Cd enrichment and resistance caused by DA; (iii) to conduct a preliminary exploration of the DA signaling pathway in plants; (iv) to study the change of endogenous DA content in duckweed during Cd stress and reveal its mechanism by analyzing DA metabolic pathway.

## 2. Results

### 2.1. Effects of DA on the Frond and Root of Duckweed under Cd Stress

Exogenous DA was applied to study the effect of DA on duckweed with Cd stress. The results in Figure 1 show that DA addition significantly improved the resistance to Cd stress in duckweed. As shown in Figure 1a, the chlorisis degree of fronds decreased by DA addition. In Figure 1b, the root abscission rate was 0% without Cd (CK), and the root abscission rate was 21.1%, 20.7%, 10.4%, 6.6%, 6.2%, and 7.9% with 0, 10, 20, 50, 100, and 200 μM DA addition under Cd stress, respectively. Root abscission rate was clearly decreased by exogenous DA, and the lowest root abscission rate was observed at the concentrations of 50 μM and 100 μM. It could also be observed in Figure 1c that 50 μM DA addition decreased the root abscission rate significantly. Therefore, exogenous DA enhanced duckweed’s Cd resistance. As a result, 50 μM DA was adopted for the following studies.

### 2.2. Chlorophyll Content and Photosynthesis of Duckweed Were Improved by DA under Cd Stress

It was found that DA addition recovered the chlorophyll content of duckweed under Cd stress. As shown in Figure 2c, Cd stress caused a decrease in Chla, Chlb, and Chl, with 0.376, 0.682, and 0.080 mg/g, respectively. However, they were elevated by DA addition; Chla, Chlb, and Chl were 0.470, 0.853, and 0.116 mg/g, respectively, 25%, 25%, and 45% higher than during Cd stress.

Furthermore, photosynthesis was measured with DA addition during Cd stress. Shown as Figure 2b, Cd decreased Fv/Fm, Y(I), Y(II), and qP, while causing an increase in Y(ND). Under Cd stress, Fv/Fm, Y(I), Y(II), qP, and Y(ND) were 0.690, 0.551, 0.501, 0.800, and 0.391, respectively. However, DA addition increased photosynthesis capacity in duckweed during Cd stress. With DA addition, Fv/Fm, Y(I), Y(II), and qP were 0.753, 0.659, 0.590, and 0.842, respectively, 9%, 20%, 18%, and 5% higher than during Cd stress. Meanwhile, Y(ND) decreased to 0.264, 32% lower than during Cd stress. Furthermore, the expression of genes related to photosynthesis and photosynthesis-antenna proteins (Table 1) was studied. The results show that exogenous DA significantly enhanced the activity and stability of PS II and PS I, as well as the concentration of antenna proteins. Moreover, there was no significant difference in the gene expression between the duckweed treated with DA alone and Control (CK) (Appendix A).

### 2.3. Cd Accumulation Increased under DA Treatment

Cd content in duckweed tissues was investigated to clarify the impact of DA treatment on the capacity of Cd absorption. As seen in Figure 3a, after 48 h of Cd and DA co-treatment, the Cd content in duckweed was 442.41 mg/kg, which was 30% higher than that in Cd group (341.37 mg/kg). 

Moreover, to further investigate the Cd content in the duckweed, protoplasts from the roots were harvested for flow cytometric analysis (Figure 3b and Appendix A). The Leadamium signal has been shown by the intensity of Ch02, and the mean of the Leadamium signal intensity (marked by green line) in more than 3000 protoplasts was significantly different in the group of DA-Cd when compared with that of Cd group. In the DA-Cd group, the mean of the Leadamium signal intensity was 2325.91, and that in the Cd group was 1655.32. Shown in the column diagram in Figure 3b, the column chart result was the average of six repetitions, and we found that there was a significant difference between these two groups.

Furthermore, the Cd^2+^ flux was measured by NMT after immersion for 30 min in a test solution. It was found that the higher Cd content in duckweed was caused by the faster Cd^2+^ influx rate of roots. According to Figure 3c, the Cd^2+^ flux cannot be measured without Cd addition. The Cd addition at the point of 5 min and net Cd^2+^ influx of 208.72 (6 min) to 101.50 pmol∙cm^−2^·s^−1^ (12 min) was measured, and it maintained 86.84 pmol∙cm^−2^·s^−1^ (14 min) in the Cd group. As regards the DA-Cd group, a net Cd^2+^ influx of 107.55 (6 min) to 125.04 pmol∙cm^−2^·s^−1^ was measured, and it maintained 115.14 pmol∙cm^−2^·s^−1^ at 14 min. After Cd exposure for 30 min, a net Cd^2+^ influx of 36.49 to 13.22 pmol∙cm^−2^·s^−1^ was measured from the beginning to 180 s. However, the influx remained at a relatively high level, about 40.09 to 27.88 pmol∙cm^−2^·s^−1^, at the same time. Subsequently, the Cd^2+^ influx increased to 30.5 and finally decreased to 15.46 from 432 s to 756 s in duckweed with Cd treatment. Compared with that, the Cd^2+^ influx sustained a steady high rate at about 27.10 pmol∙cm^−2^·s^−1^ at that time with DA addition. These results together show that Cd accumulation increased under DA treatment.

### 2.4. DA Recovered Ca^2+^ Content by Mediating Ca^2+^ Flux under Cd Stress

Ca^2+^ homeostasis is essential for plants. To clarify the mechanism of DA in plants, the Ca^2+^ content in cells was measured. As shown in Figure 4a, Cd stress decreased Ca^2+^ fluorescence intensity (1114), compared with an intensity of 1367 without Cd stress. However, under DA addition, the Ca^2+^ fluorescence intensity was elevated (1400), with an increase of 21%.

NMT was used to explore the Ca^2+^ flux under Cd stress with or without DA. A relatively stable Ca^2+^ influx was found in the CK group, with 48.41 to 12.78 pmol·cm^−2^·s^−1^. However, under Cd stress the direction of Ca^2+^ flux was changed, with a fast efflux of 93.45 to 58.49 pmol·cm^−2^·s^−1^. Compared with that, DA addition alleviated this fast efflux slightly, with an efflux of 87.81 to 34.64 pmol·cm^−2^·s^−1^.

### 2.5. DA Addition Promoted Proteins Related to Cd Resistance and Accumulation

As shown in Figure 5, transcriptome analysis was used to trace the mechanism of DA-induced Cd enrichment and detoxification. First, the copper transporter (COPT1), nitrate transporter (NRT1), which are located in cell membrane, had their expression enhanced by DA addition, with an increase by 1.31, and 1.29 log_2_ Fold Change, respectively. Second, DA addition increased the expression of the multidrug resistance protein (MRP) and vacuolar iron transporter 1 (VIT1), which are located in the vesicle membrane, with an increase by 1.5 and 4.8 log_2_ Fold Change, respectively. Third, the expression of glutathione S-transferase (GST) was enhanced by DA addition, with 1.25 log_2_ Fold Change. Fourth, in terms of metabolism, DA addition expedited the Calvin cycle, TCA cycle, and glycolysis. Among these, as a result of an increase of 3.10 log_2_ Fold Change of Rubisco’s expression, the Calvin cycle was the most up-regulated process. Moreover, the increased expression by 1.06 log_2_ Fold Change of alcohol dehydrogenase (ADH) was noteworthy. The different expression genes (DEGs) related to Cd resistance and accumulation, in the case of “DA-Cd vs Cd”, have been listed in Appendix A.

### 2.6. A Possible Site of DA Targets in Plants Is CNGC2

To illustrate possible DA targets in plants, a hypothesis map of DA signal was firstly built in Figure 6a. It was found that, under Cd stress, the dual messenger system was strengthened, and the expression of PLC and IP3R were increased by 2.69 and 3.40 log_2_ Fold Change, respectively. However, meanwhile, the ADCY-CNGC2 pathway was weakened and the expression of CNGC2 was decreased by 5.37 log_2_ Fold Change. This may be one of the reasons for the rapid efflux of Ca^2+^ caused by Cd. More interestingly, with DA addition as shown in Figure 6b, the ADCY-CNGC2 pathway was strengthened and the expression of CNGC2 and ADCY were increased by 2.00 and 0.40 log_2_ Fold Change, respectively, which would restore Ca^2+^ influx and thus alleviate the fast outflux of Ca^2+^. As the aforementioned finding was in strong agreement with results measured by flow cytometry and NMT, CNGC2 may be the target of DA in plants. All DEGs related to DA signal, in the case of “Cd vs CK” (Appendix A) and “DA-Cd vs Cd” (Appendix A), have been listed.

### 2.7. Endogenous DA Content Decreased under Cd Stress

In addition to exogenous DA application, the endogenous DA of duckweed under Cd stress was studied. According to the ELISA findings as shown in Figure 7, the DA content was 1.80 μg/kg after 24 h Cd exposure, almost 17% less than the control group’s content of 2.17 μg/kg. This result demonstrates that Cd stress decreased DA content in duckweed.

### 2.8. DA Biosynthesis Pathway Was Inhibited by Cd Stress

The expression of genes related to the DA metabolism pathway was studied in Figure 8, and the effect of Cd stress on the pathway was revealed. It was found that chorismate mutase (CM) and shikimate kinase (SK), two enzymes in the shikimate acid pathway, significantly down-regulated their gene expression, which decreased by 2.38 and 3.43 log_2_ Fold Change, respectively. This would block the production of the DA precursors’ tyrosine (L-tyr). L-tyr is produced by Aspartate aminotransferase (AST) and Tyrosine aminotransferase (TAT). While there was mixed regulation of AST, the significant down-regulation of the TAT gene, with a decrease by 3.03 log_2_ Fold Change, would further the reduction in Tyr levels. Finally, the lower level of Tyr caused a lower content of DA in duckweed under Cd stress. All DEGs related to the DA biosynthesis pathway in the case of “Cd vs CK” have been listed in Appendix A.

## 3. Discussion

Animals need quick, extensive molecular communication structures to coordinate sensory and physiological reactions all across their bodies, and DA plays an essential role in these activities. Since plants are unable to move actively like animals, their systems for coping with external challenges are more likely to be mirrored in special intracellular life activities, which have not been explored thoroughly, and DA is one of these [6]. DA has been shown to play a significant role in processes of not only salinity stress [27] and nutrient stress [13], but also drought stress [21] and organic pollution [28]. However, there has not been any research on DA in the context of heavy metal stress. The current work seeks to improve the understanding of DA in plants’ Cd stress response and to propose new research methods and tools to increase phytoremediation capacity. In this study, we discovered that even Cd stress reduced numerous growth indices, and exogenous DA supplementation not only effectively mitigated this harm, but also significantly increased the Cd enrichment capacity of duckweed. This research also discusses probable DA signaling pathways in plants and the mechanisms of DA-induced Cd enrichment.

Photosynthesis, which is how plants obtain their carbon and energy, is very sensitive to environmental stress and thus is one of the hallmarks of stress on plants. Stress ruins chloroplast architecture, impedes electron transport, harms pigment complexes, and lowers photosynthetic rates [21,29,30]. Researchers from Northwest A&F University have discovered that giving *Malus hupehensis* 100 or 200 μM of DA prevented the reduction of Fv/Fm and chlorophyll content brought on by salt stress. Additionally, the administration of 100 μM DA increased photosynthesis capacity and chlorophyll levels in *Malus hupehensis* under nutritional and drought stress [11,13,27]. Furthermore, by applying 50 mM and 100 μM DA, the same photosynthetic restoration function was obtained in organic matter pollution, such as hydrocarbon and bisphenol A [15,28]. First, this study investigated the recovery of photosynthesis capacity by DA addition in duckweed under Cd stress. It was discovered that exogenous DA markedly increased chlorophyll content and chlorophyll fluorescence characteristics (Figure 2), which means that DA protects plants from losing their green colour and allows them to maintain basic photosynthetic capacity under Cd stress.

Furthermore, to figure out which proteins were affected by exogenous DA and, thus, to enhance photosynthesis, DEGs within photosynthesis and photosynthesis-antenna proteins (Table 1) were analyzed and arranged. The expression of a total of seven proteins was increased. Among them, psbR plays an important role in the oxygen-evolving complex [31] and resistance to abiotic stresses [32]. Moreover, the strengthened expression of PsbE stabilizes the structure of the PSII electron transport chain. Furthermore, increased expression of PsaO could better balance the energy transfer between PS I and PS II [33]. Moreover, psaE, on the reducing side of PS I, plays a major role in avoiding the leakage of electrons to oxygen in the light (Mellor reaction) and consequently inhibits the formation of toxic oxygen substances [34]. The above results pointed out that exogenous DA significantly enhanced the activity and stability of PS II and PS I, as well as energy utilization and electron transfer efficiency. Moreover, DA addition enhanced the expression of Lhca3, Lhcb1, and Lhcb6, indicating the recovery of chlorophyll content and enhancement of light energy capture under Cd stress. Therefore, DA addition dramatically mitigated the damage to photosynthetic systems’ exposure to Cd stress.

Avoidance and tolerance are two ways to build plant resistance to the heavy metals [35]. Thus, by preventing the uptake and translocation of Cd^2+^, plants can improve their Cd resistance [36,37]. In addition, it has also been established that plants can demonstrate greater Cd resistance while expanding their Cd enrichment capacity [38]. In the current study, we discovered that exogenous DA increased the Cd enrichment capacity of duckweed by 1.3 fold, and this was due to a notable increase in the rate of Cd uptake by the roots of duckweed, according to results from NMT and Cd fluorescence analysis (Figure 3). Numerous ions, including Fe^2+^, Ca^2+^, Cu^2+^, Zn^2+^, and Mn^2+^, are necessary for plant development. Due to the comparable physical and chemical properties of Cd^2+^, Cd^2+^ can share ion channels with these ions and thus be absorbed and transported [39,40,41,42]. Meanwhile, one of the major ways for plants to regulate hazardous substances, such as Cd, is by vacuolar sequestration [43]. Located in the tonoplast, by transporting Fe^2+^ in cytoplasm to vacuoles, VIT1 maintains iron homeostasis in plants [44]. The possible Cd transport function of the VIT1 protein in diatoms was predicted by Tore et al. in 2011 [45], however, this function was not further investigated over the next 10 years. In this study, we discovered that DA treatment increased the expression of VIT1 by 4.8 log_2_ Fold Change, which might significantly contribute to the Cd enrichment of plants while reducing the toxicity brought on by Cd. As a result, it was hypothesized that the VIT1 protein is probably the primary reason why duckweed accumulates Cd and becomes tolerant to it; this is caused by DA. Moreover, GST catalyzes intracellular Cd detoxification reactions by first forming a cell membrane conjugate between the thiol peptide glutathione (GSH) and Cd, and then sequestering this conjugate (GS conjugate) into the vacuolar region of plant cells through MRP [46,47]. Therefore, another possible explanation for Cd resistance is the considerable upregulation of GST and MRP expression, elevated by DA in our research. Additionally, the NRT1 family regulates NO_3_^−^ allocation in roots to organizing Cd^2+^ accumulation in root vacuoles, which also encourages Cd^2+^ detoxification [48,49,50]. In this investigation, we discovered that the higher Cd resistance in duckweed may possibly be a result of the DA-induced increase in NRT1 expression.

Additionally, a metabolic perspective on Cd detoxification analysis was made. Studies have reported that overexpression of ADH increased the transcript levels of multiple stress-related genes, accumulation of soluble sugars, and callose depositions, indicating that ADH conferred enhanced resistance to both biotic and abiotic stresses [51]. This may be one of the reasons why DA enhances plant stress resistance. Moreover, Rubisco (RBC) is the key enzyme that determines the carbon assimilation rates in photosynthesis, as well as an indispensable oxygenase in photorespiration. One of the symptoms of Cd stress is the reduced expression of Rubisco in plants, which negatively affects photosynthesis [52]. However, increased expression of Rubisco by 3.10 log_2_ Fold Change caused by DA addition under Cd stress was observed in this study, which is consistent with the conclusion that DA can improve photosynthesis. 

As early as 2005, studies on DA signaling in plants were reported. Using cDNA expressing human DA receptors to modify potato plants, Skirycz et al. discovered the increased activity of starch phosphorylase, but the decreased activity of ADPglucose pyrophosphorylase, causing an increased level of sucrose and a decreased level of starch. As a result, they hypothesized that DA activates cAMP signaling in potato, which in turn controls how plants use carbon [22]. More interestingly, higher sucrose accumulation but lower starch levels caused by DA pretreat in apple were also reported in 2020 [21], which means it is probably a common phenomenon in plants. According to transcriptome analyses, Gao et al. reported that DA may improve apple’s ability to resist drought by boosting the expression of genes from the CNGC and CAM/CML families. The present findings successfully combine earlier research. Based on the results of Ca^2+^ fluorescence density, NMT measurement (Figure 4), and transcriptome analyses (Figure 6), it was hypothesized that in plants, DA activated ADCY, which activated CNGC2 by cAMP to restore the intracellular Ca^2+^ concentration, and then a number of downstream signals mediated by cAMP and Ca^2+^ were elevated, such as PKA and CBL/CAM-dependent signaling pathways, producing a series of cytological effects that ultimately manifest as Cd enrichment and resistance. As a result, an upstream location for DA targets in plants may be CNGC2 caused by cAMP.

As well as exogenous DA application, endogenous DA content was also studied here. With the activity of TYDC, DD, or TH being increased under high salt stress, UV irradiation, and drought stress, while inhibited under low temperature, darkness, and red light treatment, the impact of various stresses on endogenous DA production in plants appears to be specific [10]. Alkaline stress has also been reported to increase DA levels in plants [12]. In this work, we discovered that Cd stress blocked the shikimate acid pathway by decreasing the expression of SK and CM and, thus, reduced the level of DA precursor, Tyr. Therefore, DA level was then also decreased (Figure 8). This finding also corresponds with the result measured by the ELISA test (Figure 7).

## 4. Materials and Methods

### 4.1. Plant and Tissue Culture Conditions

Duckweed (*Lemna turionifera* 5511) was collected and cultured according to our previous study [53]. It was cultivated with a light intensity of 95 μmol m^−2^·s^−1^ and a light duration of 16 h per day [54], while the daytime and overnight temperatures were 26 °C and 20 °C, respectively [55].

### 4.2. DA Treatment Concentration under Cd Stress

The duckweed was given a 24 h pretreatment in the culture medium containing 0, 10, 20, 50, 100, and 200 μM DA before being moved to a datko containing 0, 10, 20, 50, 100, and 200 μM DA, respectively, with or without 50 μM CdCl_2_. According to related studies and our previous work [53,55,56], 50 μM CdCl_2_ is an appropriate strength of Cd stress in *Lemna turionifera*. In addition, 50 μM CdCl_2_ was used to treat samples in experiments on growth, Cd accumulation, Cd^2+^ flux, and transcriptome analysis. The root abscission rate was measured after 24 h, and the duckweed phenotype was observed after 48 h. The treatment groups were: (1) control treatment, cultured in liquid medium (CK group); (2) DA treatment, cultured in the liquid medium added with 50 μM DA; (3) Cd treatment, cultured in the liquid medium added with 50 μM CdCl_2_ (Cd group); (4) DA and Cd treatment, cultured in the liquid medium added with 50 μM DA and 50 μM CdCl_2_ (DA-Cd group) [13,27,57]. The chlorophyll fluorescence, protoplast study, DA content, and RNA sequencing was measured after 24 h treatment. The duckweed phenotype and chlorophyll content was observed after 48 h. 

### 4.3. Measurement of Chlorophyll Fluorescence

The Dual-PAM100 fluorometer from Waltz, Germany, was used to measure the fluorescence of chlorophyll. The samples were given a 30-min dark treatment prior to the measuring in order to reduce the amount of organic materials in the plant tissue before detection. The maximum quantum efficiency of photosystem II (Fv/Fm), the actual quantum efficiency of photosystem II [Y(II)], photochemical quenching coefficient (qP), photochemical quantum efficiency of photosystem I [Y(I)], and the non-photochemical energy dissipation due to the donor-side limitation [Y(ND)] were detected.

### 4.4. Measurement of Chlorophyll Content

A total of 0.2 g duckweed (FW) was harvested and soaked in 30 mL 95% alcohol for 48 h. After that, chlorophyll extract solution was added to 96-well plates and the absorbance was measured at 663 nm and 645 nm with a microplate reader (TECAN, Infinite M200 Pro, Seestrasse, Männedorf, Switzerland) [8].

### 4.5. DA Content Measurements

The DA content was measured by DA assay kit (Enzyme-linked Biotechnology Co., Shanghai, China). A total of 0.1 g Duckweed (FW) was thoroughly ground under liquid nitrogen and extracted in 1 mL of 0.01M PBS. The extract was applied to an ELISA plate. Each well received 100 μL of enzyme labeling reagent (HRP), which was then incubated at 37 °C for 1 h. After that, the liquid was drained from the plate and thoroughly washed in washing solution. The chromogenic reaction was carried out at 37 °C for 15 min after 50 μL of substrates A and B were added to each well to begin the reaction. After adding 50 μL of termination solution, the OD value was measured at 450 nm by a microplate reader (TECAN, Infinite M200 Pro) within 15 min. The absorbance of the standards was used to draw a standard curve, and the DA content of duckweed was calculated according to the standard curve.

### 4.6. Measurement of Cd Content in Duckweed

After 48 h of treatment with or without DA under Cd treatment, the duckweed was harvested, dried in 60 °C, and nitrolysized. Then, that substance was measured by an inductive coupled plasma emission spectrometer (ICP, Agilent ICP-OES 725 ES, CA, USA) to analyze the Cd concentration in duckweed.

### 4.7. Flow Cytometric Analysis of Intracellular Ca^2+^ and Cd^2+^ Content

The protoplasts were collected in the duckweed with CK, Cd, and DA-Cd treatment, following our previous study [52]. First, 95% ethanol was used to fix the treated duckweed for 10 min. The fronds and roots of the duckweed were divided, and the roots were used for the following treatments. They were then incubated for 60 min at 37 °C in the dark with 1% cellulase and 1% pectinase to produce protoplasts, which were subsequently cleaned three times with DPBS. After that, they were stained in the dark with 30 μL Leadamium^TM^ Green AM dye at 37 °C for 90 min or 30 μL Fluorescence-4 AM at 37 °C for 60 min. They were washed three times with DPBS, and then put through a 400-mesh cell strainer. Finally, the protoplasts were analyzed. The sample counts for each test were more than 3000, and each group had 6 parallel samples. The fluorescence intensity was measured with a Flow Sight (Merck millipore, FlowSight^®^ Imaging Flow Cytometer, Darmstadt, Germany). The different channels show the different identifications motivated by different lasers at different wavelengths, shown in Appendix A.

### 4.8. Measurement of Cd^2+^ and Ca^2+^ Flux

Non-invasive Micro-test Technology was employed (NMT Physiolyzer^®^, Younger, Amherst, MA, USA; Xuyue Company, Beijing, China) to measure Cd^2+^ and Ca^2+^ flux in real time. The NMT test solution was 0.1 mM CaCl_2_, 0.1 mM KCl, 0.3 mM MES, 50 μM CdCl_2_, and with or without 50 μM DA (pH 5.5). The roots were placed in the Petri dish with the relevant test solution and immersed for 30 min. The flux microsensor was positioned around 10 µm from the root surface and began to detect at the root surface location, which was 100 µm from the root tip. Each site was tested for 10 min [52,58,59]. The controls (without duckweed under Cd stress) for flux rate text are shown in Appendix A, and the Cd flux of artificial Cd^2+^ absorbing Cd source and blank are shown in Appendix A. Besides, another control has also been set and shown in Appendix A, in which, the Cd^2+^ flux of artificial Cd^2+^ absorbing source and blank has also been measured.

### 4.9. RNA Sequencing

After 24 h of treatment under the appropriate conditions, the duckweed from the CK, Cd, and DA-Cd groups was collected and sequenced at Novogene (Chaoyang, Beijing, China). Firstly, the asparagine polysaccharide polyphenol kit (QIAGEN, Hilden, Germany) was used to obtain the total RNA of the duckweed and then the integrity and the total quantities of RNA were determined by an Agilent 2100 bioanalyzer. Secondly, Oligo (dT) magnetic beads were used to enrich the mRNA with polyA tails before they were transported to the fragmentation buffer, whose divalent cation could randomly cut the whole mRNA. Using fragmented mRNA as a template and random oligonucleotides as primers, the first strand of cDNA was created in the M-MuLV Reverse Transcriptase System. The RNA strand was then degraded in the presence of RNaseH. Using dNTPs as the starting material, DNA polymerase I created the second strand of cDNA. The purified double strand underwent end repairs, an A-tail was inserted, and a sequencing adaptor was fastened. The cDNA with a length of 370–420 bp was then chosen and purified using AMPure XP beads (Beckman Coulter, Beverly, MA, USA). The library was acquired after the PCR product underwent a second round of purification. The library was then diluted to 1.5 ng/μL and tested using qRT-PCR to make sure its effective concentration was larger than 2 nM. Thirdly, Illumina NovaSeq 6000 sequencing was carried out using the theory of Sequencing by Synthesis after library qualification was verified. The sequenced flow cell had four fluorescently tagged dNTPs, DNA polymerase, and splice primers added for amplification; this allowed researchers to capture and analyze the freshly added dNTP’s fluorescence in order to determine the fragment’s sequencing.

### 4.10. Statistical Analysis

At least three distinct biological replicates were used to set up all of the data collected during the studies. The difference of root abscission was tested by LSD method. One-way ANOVA with turkey method was employed to examine the differences between three or more groups. An independent samples t-test was employed to examine the differences between two groups. Microsoft Excel 2019 was used to store the data, GraphPad Prism 8 to plot it, and Adobe Illustrator 2020 to create the pattern illustrations. GraphPad Prism 8 was also used to determine whether the results were statistically different. Standard deviations are shown as error bars (SD). Significant differences are shown by asterisks (* *p* < 0.05, ** *p* < 0.01).

## 5. Conclusions

In summary, our findings assess the effect of DA on plants under Cd stress. Physiological analyses and RNA-Seq were used to show that DA could alleviate damage on duckweed caused by Cd stress. Firstly, DA not only decreased root abscission rate, but also increased photosynthetic capacity and chlorophyll content, which was the result of increased expression of genes related to PS I and PS II, as well as the LHC family. Furthermore, DA addition could enhance the capacity of duckweed to remediate Cd pollution. It was found that the Cd^2+^ influx rate was accelerated by DA, making Cd content 30% more than CK. Transcriptome analyses revealed the mechanism, which was the increased expression of a series of genes related to Cd enrichment and resistance, such as VIT1, MRP, GST, NRT1, and Rubisco. Moreover, considering the results of the Ca^2+^ flux and the Ca^2+^ content of duckweed, as well as the changed expression of CNGC2 between the Cd and DA-Cd group, a possible DA target in plants is CNGC2, which is activated by ADCY and cAMP. Our findings lay the foundation for further study of the mechanism of DA in plants and provide potential for DA application in phytoremediation.

Cd stress conditions may have resulted in the response of other neurotransmitters. This could be further explored in future to improve growth and development under Cd stress.

## Figures and Tables

**Figure 1 plants-12-01996-f001:**
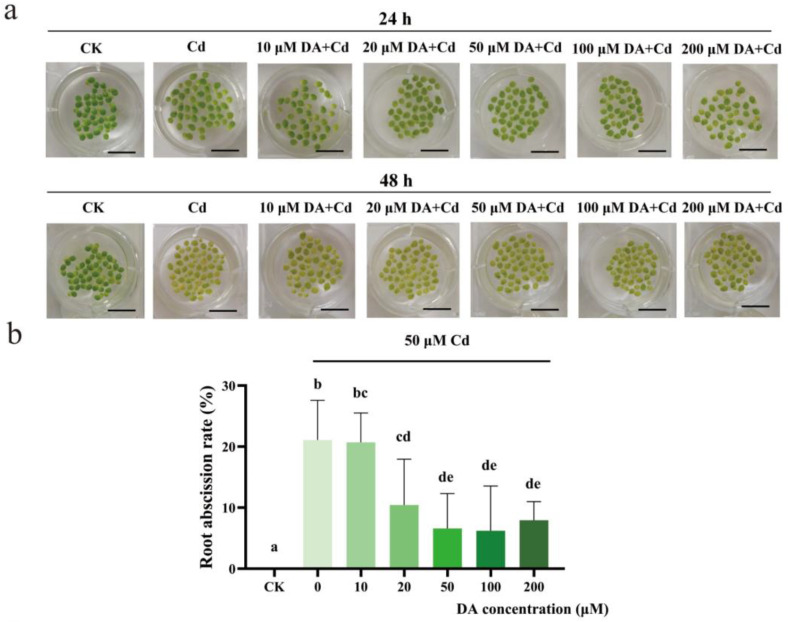
(**a**) The fronds were observed with 50 μM CdCl_2_ under different DA treatments for 24 and 48 h. Bar = 1 cm. (**b**) The root abscission rates for 24 h were measured with 50 μM CdCl_2_ under different DA treatments. LSD method was adopted and lower-case letters in the column were used to show significant differences between the groups at *p* < 0.05. (**c**) The root treated with or without 50 μM DA under 50 μM CdCl_2_ treatment for 24 h.

**Figure 2 plants-12-01996-f002:**
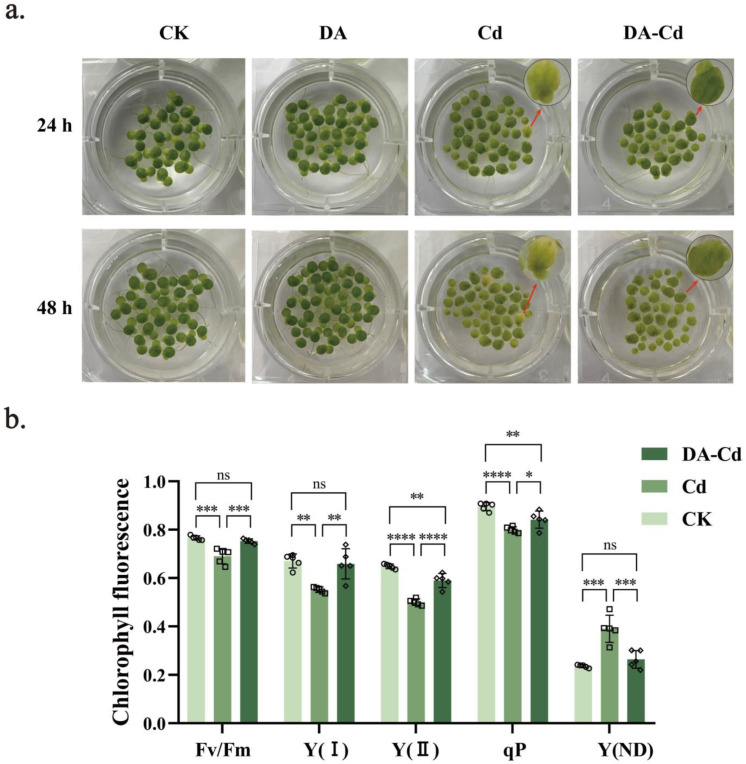
(**a**) Phenotypes of duckweed after 24 and 48 h of growth with different treatments. (**b**) The chlorophyll fluorescence Y(II), Y(I), Fv/Fm, qP, and Y(ND) were measured at 24 h. (**c**) The Chla, Chlb, Chl contents were measured at 48 h. The hollow symbols on column point exact position of every repetition. CK, only nutrient solution; DA, nutrient solution with 50 μM DA; Cd, nutrient solution with 50 μM CdCl_2_; DA-Cd, nutrient solution with 50 μM DA and 50 μM CdCl_2_. One-way ANOVA with turkey method was used. * *p* < 0.05, ** *p* < 0.01, *** *p* < 0.001, **** *p* < 0.0001, ns: not significant.

**Figure 3 plants-12-01996-f003:**
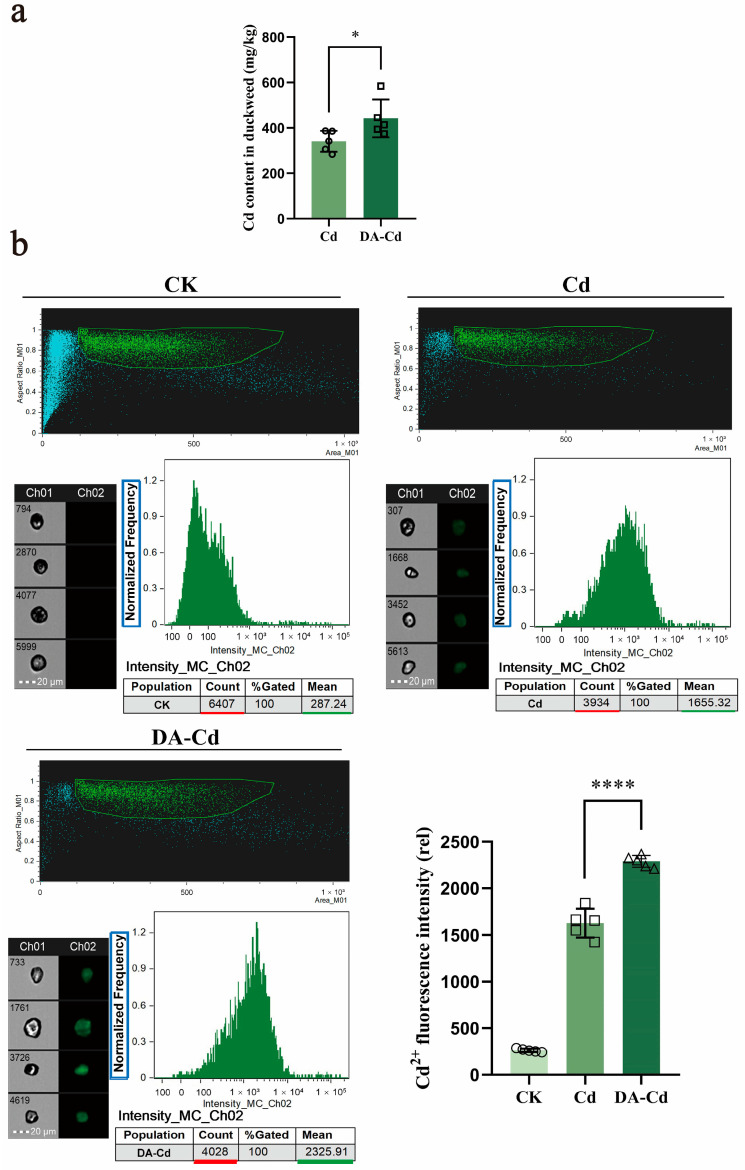
(**a**) Cd content of duckweed after 48 h exposure in 50 μM CdCl_2_ with or without 50 μM DA. (**b**) Using flow cytometry at 488 nm, Cd fluorescence of protoplast was measured. Duckweeds exposed to 50 μM CdCl_2_ were treated with or without 50 μM DA for 24 h. Then, protoplasts were harvested and dyed with Leadamium^TM^ Green AM. The bright field is Ch 01, while the 488 nm excitation light is Ch 02, with the scale bar of 20 μm. The protoplasts, framed in green, were selected to analyze fluorescence intensity. The sample amount has been marked by red lines, and the mean of the intensity has been marked by green lines. One-way ANOVA with turkey method was used. * *p* < 0.05, **** *p* < 0.0001. The hollow symbols on column point exact position of every repetition. (**c**) NMT was used to measure the net Cd^2+^ flux between 0 to 15 and 30 to 40 min. Duckweed was exposed to Cd stress at 5 min. Cd, test solution with 50 μM CdCl_2_; DA-Cd, test solution with 50 μM DA and 50 μM CdCl_2_.

**Figure 4 plants-12-01996-f004:**
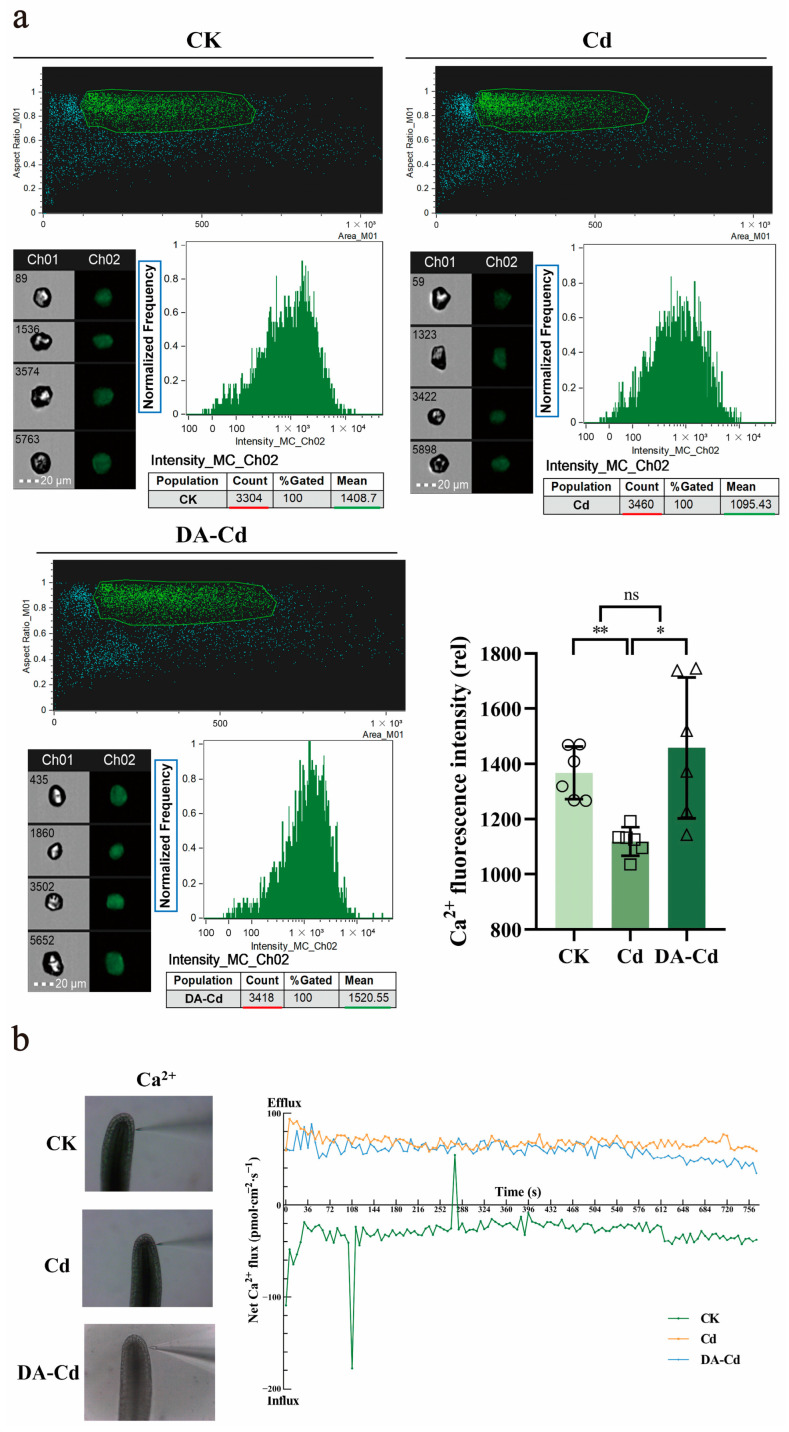
(**a**) Using flow cytometry, the protoplast’s Ca^2+^ fluorescence intensity was examined. Exposed to 50 μM Cd, duckweed was treated with or without 50 μM DA for 24 h. After that, Fluorescence-4 AM was used to dye their protoplasts. The bright field is Ch 01, while the 488 nm excitation light is Ch 02, with the scale bar of 20 μm. The protoplasts, framed in green, were selected to analyze fluorescence intensity. One-way ANOVA with turkey method was used. * *p* < 0.05, ** *p* < 0.01, ns: not significant. (**b**) Micrographs of the Ca^2+^ flux measurement at 100 μm from the root tip of duckweed. CK, only test solution; Cd, test solution with 50 μM CdCl_2_; DA-Cd, test solution with 50 μM DA and 50 μM CdCl_2_. NMT was used to measure the net Ca^2+^ flux.

**Figure 5 plants-12-01996-f005:**
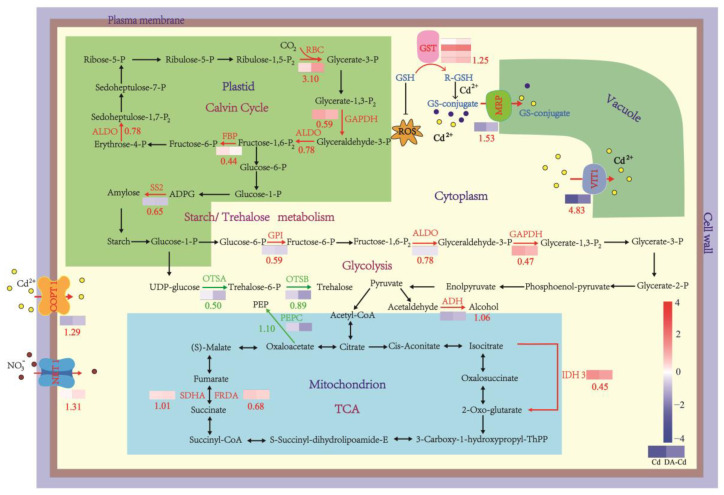
Mechanism of Cd detoxification and accumulation caused by DA in *Lemna turionifera* 5511. The catalytic processes or transit directions are indicated by the arrows. Red represents up-regulation, while green represents down-regulation, and the values beside indicate the change of DEGs corresponding to a protein. Heatmaps were adopted to display the expression pattern of DEGs, and red to blue meant that the expression level was from high to low. Yellow and blue orbs represent free Cd^2+^ and chelated Cd^2+^, respectively. Abbreviations: VIT1, vacuolar iron transporter 1; RBC, ribulose-bisphosphate carboxylase; GST, glutathione S-transferase; NRT1, nitrate transporter 1; COPT1, copper transporter 1; MRP, multi-drug resistance-associated protein; ADH, alcohol dehydrogenase; SDHA, succinate dehydro-genase subunit 8A; ALDO, fructose-bisphosphate aldolase; FRDA, fumarate reductase flavo-protein; SS2, starch synthase II; GPI, glucosephosphate isomerase; GAPDH, NADP-dependent glyceraldehyde-3-phosphate dehydrogenase-like; IDH3, isocitrate dehydrogenase 3 (NAD^+^); FBP, fructose-1,6-bisphosphatase; OSTA, trehalose 6-phosphate synthase; OSTB, trehalose 6-phosphate phosphatase; PEPC, phosphoenolpyruvate carboxykinase.

**Figure 6 plants-12-01996-f006:**
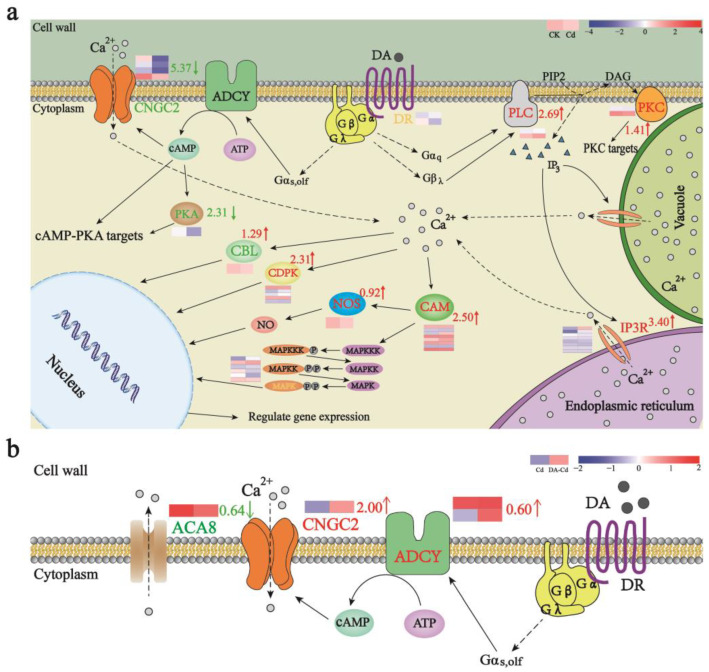
Hypothesis map of DA signal in plants. (**a**) Effect of Cd stress on DA signaling pathway. (**b**) In the presence of Cd, the effect of DA addition on DA signaling. Green represents down-regulation, while red represents up-regulation, and the values beside indicate the average change of all DEGs corresponding to a protein. Cluster heatmaps were plotted by normalization [log_2_ (FPKM+1)] and standardization (Z-score), so the discrepancy of unigenes expression between both treatments could be demonstrated. Red means high expression, while blue means low expression. The solid cone arrows indicate activation, while the dashed cone arrows indicate transport. Abbreviations: CNGC2, cyclic nucleotide-gated ion channel 2; PKA, cAMP-dependent protein kinase; ADCY, adenylate cyclase; DR, dopamine receptor; PLC, phospholipase C; PKC, protein kinase C; IP3R, IP3 receptor; CAM, calmodulin; MAPKKK, mitogen-activated protein kinase kinase kinase; MAPKK, mitogen-activated protein kinase kinase; MAPK, mitogen-activated protein kinase; NOS, nitric oxide synthase; CDPK, calcium dependent protein kinase; CBL, calcineurin B-like proteins; cAMP, cyclic adenosine monophosphate; DA, dopamine; PIP2, phosphatidylinositol-4,5-bisphosphate; DAG, diacylglycerol; IP3, inositol triphosphate; NO, nitric oxide.

**Figure 7 plants-12-01996-f007:**
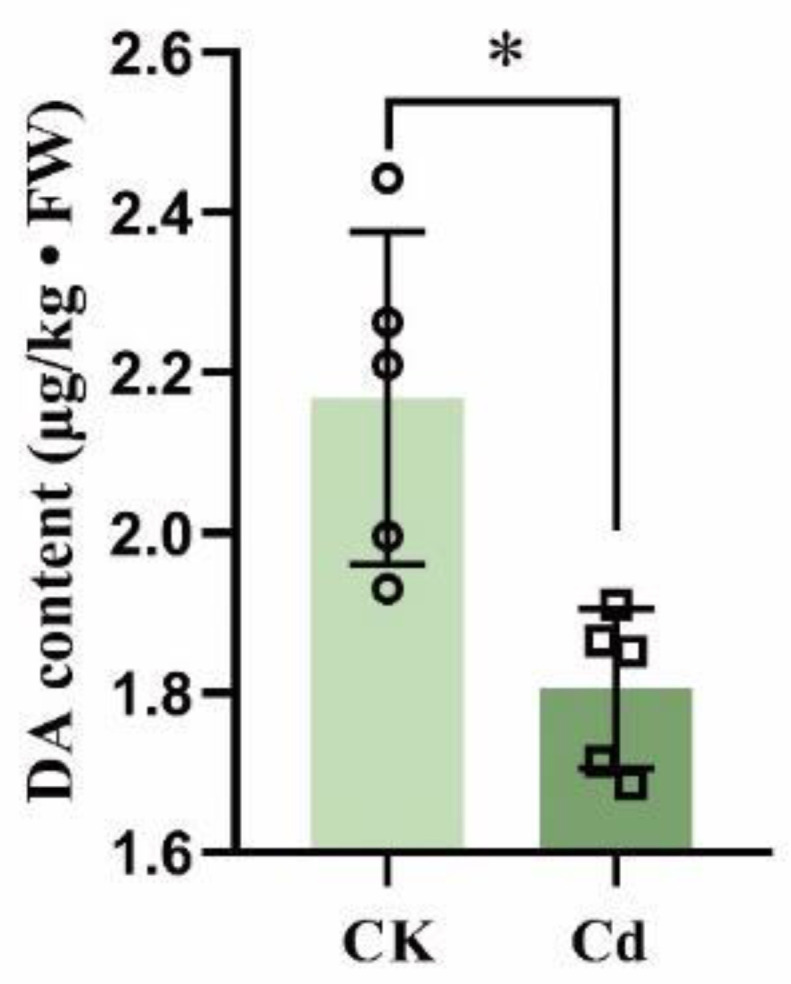
Endogenous DA level of duckweed with or without 24 h Cd exposure. With five biological repeats, the hollow symbols on column point exact position of every repetition, an independent samples *t*-test was employed to examine significant differences, which are denoted by asterisks (* *p* < 0.05).

**Figure 8 plants-12-01996-f008:**
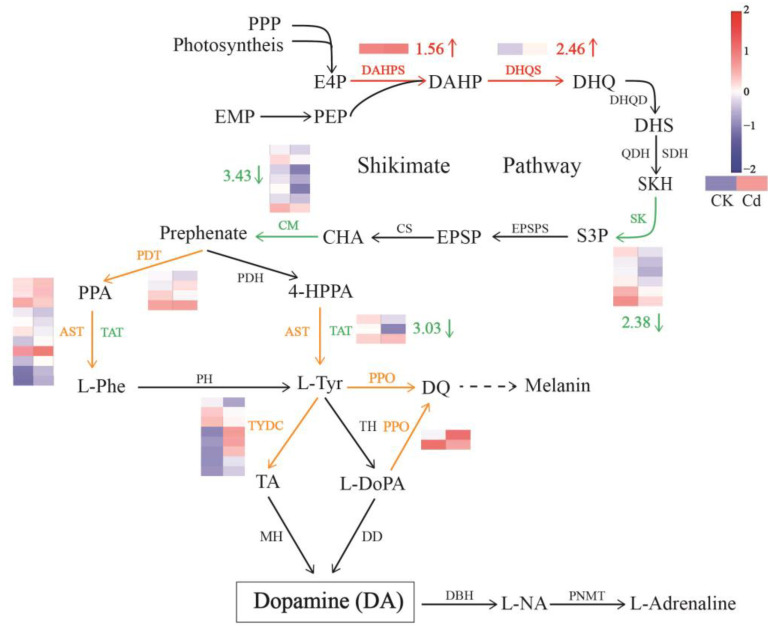
Response of DA metabolic pathway to Cd stress in duckweed. Using normalization [log_2_ (FPKM+1)] and standardization (Z-score), cluster heatmaps were created to demonstrate the variations in unigene expression between the two groups. Colors ranging from red to blue denote high to low expression. The expression of the genes dictates the color of an arrow. Red and green denote increased and decreased regulation, respectively, while yellow denotes mixed regulation. The arrow’s value indicates the average change in DEGs’ expression. Abbreviations: PPP, pentose phosphate pathway; EMP, glycolysis; PEP, phosphoenolpyruvate; E4P, erythrose 4-phosphate; DAHP, 3-deoxy-d-arabino-heptulosanate-7-phosphate; DHQ, 3-dehydroquinate; DHS, 3-dehydro shikimate; SKH, shikimate; S3P, shikimate 3-phosphate; EPSP, 5-enolpyruvylshikimate 3-phosphate; CHA, chorismite; PPA, phenylpyruvic acid; L-phe, L-phenylalanine; 4-HPPA, 4-hydroxyphenylpyruvic acid; L-tyr, L-tyrosine; DQ, dopaquinone; L-NA, L-noradrenaline; DAHPS, 3-deoxy-D-arabino-heptulosonate-7-phosphate synthase; DHQS, 3-dehydroquinate synthase; DHQD, 3-dehydroquinate dehydratase; QDH, quinate dehydrogenase; SDH, shikimate dehydrogenase; SK, shikimate kinase; EPSPS, 3-phosphoshikimate 1-carboxyvinyltransferase; CS, chorismate synthase; CM, chorismate mutase; PDT, prephenate dehydratase; PDH, prephenate dehydrogenase; AST, aspartate aminotransferase; TAT, tyrosine aminotransferase; PH, phenylalanine hydroxylase; PPO, polyphenol oxidase; TH, tyrosine hydroxylase; TYDC, tyrosine decarboxylase; MH, monophenol hydroxylase; DD, dopa decarboxylase; DBH, dopamine beta-monooxygenase; PNMT, phenylethanolamine N-methyltransferase.

**Table 1 plants-12-01996-t001:** Photosynthesis and antenna proteins.

Description	GeneID	DA_Cd_ Read Count	Cd_ Read Count	log2FoldChange	pval	padj
**Photosystem II 10kDa protein (psbR)**	Cluster-842.10293	2091.08	797.41	1.39	9.20 × 10^−7^	1.48 × 10^−4^
**Photosystem II 22kDa protein (psbS)**	Cluster-842.9755	6179.71	2787.42	1.15	7.87 × 10^−12^	7.21 × 10^−9^
**Photosystem I subunit IV (psaE)**	Cluster-842.8276	3329.22	1550.97	1.10	3.25 × 10^−6^	4.13 × 10^−4^
**Photosystem I subunit (PsaO)**	Cluster-842.10000	5863.39	2741.13	1.10	3.44 × 10^−8^	9.32 × 10^−6^
**Light-harvesting complex I chlorophyll a/b binding protein 3 (LHCA3)**	Cluster-842.7185	2683.81	1203.95	1.16	2.38 × 10^−8^	6.65 × 10^−6^
**Light-harvesting complex II chlorophyll a/b binding protein 1 (LHCB1)**	Cluster-842.7154	8094.49	2867.23	1.50	3.09 × 10^−11^	2.23 × 10^−8^
**Light-harvesting complex II chlorophyll a/b binding protein 6 (LHCB6)**	Cluster-842.3135	455.67	191.30	1.25	1.35 × 10^−4^	7.88 × 10^−3^

## Data Availability

The original contributions presented in the study are included in the article; further inquiries can be directed to the corresponding author.

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
