# Peer review of "New Insight into the Function of Dopamine (DA) during Cd Stress in Duckweed (Lemna turionifera 5511)"

_plants, 2023, doi:10.3390/plants12101996_

Round 1

Reviewer 1 Report

    New insight into the function of Dopamine (DA) during Cd stress in duckweed (Lemna turionifera 5511) Why did the authors target dopamine during cadmium stress in this study? What is the rationale of this study? What is the botanical and environmental relevance of the current study? Why not target gene and protein expression in this study? It will be more confirmatory. Why not target other monoamines analysis in this study? Why not targetting other signaling pathway in this study as explained in figures 6 and 8  

Reviewer 2 Report

The manuscript describes the amelioration of Cd toxicity induced by dopamine, a known neurotransmitter that is also synthetised in plants. Several previous reports showed that dopamine could help to sustain different types of stress, the hypothesis that this signalling metabolite exerts similar role under metal stress is interesting. Several measured parameters using advanced techniques showed that eventually dopamine helps to reduce the stress caused by Cd. However, additional data, in particular use and analysis of controls, are required to improve the information obtained. These are the major issues that must be solved and corrected:

1.         Include abbreviations of compounds and enzymes the first time they appear in the text, even in the abstract.

2.         It is not clear why was chosen the dose of 50 µM Cd. A series of Cd doses should have been tested, particularly to show that dopamine is already produced in duckweed in response to this toxic metal. Therefore, the first experiment to describe should have been a dose-response graph at different times of treatment, where different parameters of stress were measured: dopamine, chlorophyll, root abscission, etc.

3.         Fig. 1 says that samples were measured at 24 h and 48 h of treatment. However, this is not the case: None of the panels include different times, and it is not clear what treatments are show: The X-axis of panel seems to be a group of leaf discs or protoplasts (what is the size?) with samples exposed to Cd and different doses of dopamine. But where are the controls? Does dopamine induce root abscission? Does this mean the at higher level of dopamine more stress is detected? Please, explain. Panel B shows larger number roots floating in the media, but it is impossible to seen what doses or treatments, and, again, duckweeds under control conditions are missing.

4.         Fig. 2 shows, apparently, also data of 24 and 48 h but I cannot distinguish any relevant differences between treatments in Panel a. Why photochemical parameters correspond only to 24 h-treated plants while chlorophyll was only measured in 48 h-treated ones? Fig. 2b comprises data of control, Cd and DA-Cd. Why not DA alone?

5.         Tables 1 and 2 can be merged, and should include also data of Control and DA. What experimental conditions were used for these results? 24 or 48 h? In any case, changes seem not very high.

6.         Fig. 3. Describes data of a Cd-fluorescent probe and electrochemical analysis of Cd flow at the root tip. The flow cytometry values need to be normalised. It only shows the relative e fluorescence of the probe, but is difficult to know how much protoplasts were actually in the sample. Perhaps an option is the autofluorescence of chlorophyll, which emits in the red spectral range. Additional data must also be included: control and DA treated cells, to determine the level of background that may occur.

With regard to the electrochemical analyses, it is intriguing that there is not variation of the signal with time. It looks to be quite stable, while ideally it should change with the metabolic activity of the protoplasts. In addition, uncouplers of membrane potential and blockers of transporters and ion channels should be used to prove that the signals and the flux is genuine.

7.         Fig. 4. Similar problems happen in this figure as those commented before in Fig. 3: The flow cytometry requires normalisation, and proper controls must be added, to determine changes only in response to dopamine. This is particularly important for Ca, since this is a critical secondary messenger that may respond to the neurotransmitter. Again, appropriate uncouplers and blockers must be used during the incubation of protoplast to confirm the nature of the signal shown. It is quite intriguing that the signal does not change with time, as already commented.

8.         Maps of metabolic responses mediated by dopamine and signalling done by RNA seq. It is not clear what samples were used to detect the changes shown with the “heat” maps of the different figures.

Such hypothesis should be sustained also by confirmatory experiments focused to key components. In this sense, most of the changes refer to signalling components that are mostly regulated by post-translational mechanisms. Therefore, why transcriptional data are then relevant to understanding signalling processes and metabolic adjustments?

Additional experiments are, in consequence, required to complete the results and draw a full model to explain the dopamine signalling mechanism under Cd stress.

Reviewer 3 Report

The effects of dopamine (DA) on duckweed under the influence of cadmium stress (Lemna turionifera 5511). The findings of the current study advance our understanding of cadmium stress, which provides a fresh perspective on the use and investigation of Dopamine to the phytoremediation of cadmium in aquatic systems where DA is present. The findings are presented in a scientific manner, and there is sufficient justification for the interpretation. The writing is well organized and simple to understand. The findings are laid up in a manner that can easily be understood. The paper has multiple examples of typography as well as symbols that have been messed up. The article ought to be approved for publication provided that the minor remarks offered below are taken into consideration.

Some minor comments hereafter

  1. In Abstract the authors are advised to discuss some results.
  2. The authors are encouraged to provide a brief Graphical Abstract which can give the idea of the present work.
  3. There are many repetitive sentences which should be corrected or omitted.
  4. In Statistical analysis, it should be revise and authors should which tests were employed.
  5. Figure 5: Authors should simplify the figure or shift it to the supplementary files
  6. The authors are encouraged to add the Limitations and Future prospects of the current study after the conclusion section

Round 2

Reviewer 1 Report

As the authors addressed the reviewers' comments, I suggest acceptance of the manuscript.

Reviewer 2 Report

I am not satisfied by some responses and explanations given by the authors. I am very concerned by the data of flow cytometry and electrophysiological measurements of Cd2+ and Ca2+ fluxes. shown inf Figures 3 and 4. The authors argue that this technique was used in laurate laboratories and maple used in many biological systems. However, these arguments do not prove sufficient to merit acceptance:

Flow cytometry lacks normalization of sample amount. Despite more signal of Leadamium Green was observed in Cd treated fronds, differences with dopamine incubated duckweed seems not relevant, unless there is a counter staining or other means of fluorescence (for example chlorophyll autofluorescence could be used) to adjust the degree of Leadamium signal with equivalent protoplasts amount. These experiments could be biased by using isolated protoplasts at different concentrations.

The electrophysiological measurements done of Cd and Cd fluxes covered only an interval of 3 min, with similar results as described in Ref. 58. Which also lacks important controls, such as addition of ion channel blockers or membrane potential uncouplers.

However, Ref. 59 shows in Fig. 3 the flow of cations over a several minutes period, which eventually went up or down depending on the treatments. This is not the case of Figs. 3 and 4, which signal remained stable, while supposedly Cd2+ and Ca2+ are moving inwards or outwards with time.

In other words, the Non-invasive Micro-test Technology (NMT) used is unable to show the net flux of those cations. Of course, in the mentioned papers the Cd2+ flux became stable, but it was possible to see an evolution. If the flux becomes stable, the system is in equilibrium, and you cannot see cation transport. What you should show is the moment when Cd2+ is added and how the external concentration of Cd2+at the root tip decreases or increases with time due to influx or efflux. In addition, blockers and /or membrane potential uncouplers should be used to revert the movement, so that is feasible to measure the slope and the ratio of transport that occurs.

Nernst equation and Fick's first diffusion law are used to calculate the concentration of ions in equilibrium for a certain value of membrane potential, which was not measured in the experiment, and may be different in each sample.

The critical point here is to analyzing the flow rate of ions and molecules in your experiment, because this depends on active transport, which is directed by changes in membrane potential that may be altered, in turn, by the neurotransmitter dopamine.

Round 3

Reviewer 2 Report

Please, see the attached file with my comments.
